# D2 Receptors and Sodium Ion Channel Blockades of the Basolateral Amygdala Attenuate Lithium Chloride-Induced Conditioned Taste Aversion Applying to Cancer Chemotherapy Nausea and Vomiting

**DOI:** 10.3390/brainsci13040697

**Published:** 2023-04-21

**Authors:** Zhi-Yue Gao, Chung Ming Huang, Cai-N Cheng, Andrew Chih-Wei Huang

**Affiliations:** 1Yuanshan Branch, Taipei Veterans General Hospital, Yi-Lan County 26247, Taiwan; 2Department of Psychology, Fo Guang University, Yi-Lan 26247, Taiwan; 3Department of Life Sciences, National Central University, Jhong-Li District, Taoyuan City 32001, Taiwan

**Keywords:** LiCl, conditioned taste aversion, dopamine, D2 receptor, sodium ion channel blocker, basolateral amygdala

## Abstract

Cancer patients regularly suffer from the behavioral symptoms of chemotherapy-induced nausea and vomiting. Particularly, it is involved in Pavlovian conditioning. Lithium chloride (LiCl) was used as the unconditioned stimulus (US) and contingent with the tastant, for example, a saccharin solution (i.e., the conditioned stimulus; CS), resulted in conditioned taste aversion (CTA) to the CS intake. The present study employed an animal model of LiCl-induced CTA to imitate chemotherapy-induced nausea and vomiting symptoms. Recently, the basolateral amygdala (BLA) was shown to mediate LiCl-induced CTA learning; however, which brain mechanisms of the BLA regulate CTA by LiCl remain unknown. The present study was designed to test this issue, and 4% lidocaine or D2 blocker haloperidol were microinjected into BLA between the 0.1% saccharin solution intake and 0.15M LiCl. The results showed lidocaine microinjections into the BLA could attenuate the LiCl-induced CTA. Microinjections of haloperidol blunted the CTA learning by LiCl. Altogether, BLA via the sodium chloride ion channel and D2 receptors control LiCl-induced conditioned saccharin solution intake suppression. The findings can provide some implications and contributions to cancer chemotherapy-induced nausea and vomiting side effects, and will help to develop novel strategies to prevent the side effects of cancer chemotherapy.

## 1. Introduction

Chemotherapy for cancer treatment has been used for over 50 years. Although antiemetic drugs have been developed to reduce chemotherapy’s cytotoxic drug-induced side effects, approximately 25–30% of patients often have severe side effects, including anticipatory nausea and vomiting [1]. Taste aversion (e.g., nausea and vomiting) occurs soon following chemotherapy. Taste aversion is often achieved in a single-trial type of classical conditioning, and within 24 h after the first course of chemotherapy [2,3]. Such taste aversion has been explained by a specific type of Pavlovian conditioning, conditioned taste aversion (CTA), in which daily tastants (conditioned stimuli [CSs]) are conditioned to the subsequent side effects of the drugs (e.g., cytotoxic drugs or unconditioned stimuli [USs]) used to treat the illness [4]. Some clinical studies have shown that cancer patients who undergo repeated chemotherapy exhibit an increase in the strength of their nausea and vomiting [5,6,7,8].

Concerning studies of chemotherapy nausea and vomiting symptoms, animal models of LiCl-induced CTA learning and produced CS suppression can be applied to imitate nausea and vomiting behaviors [3,9,10]. Moreover, a crucial issue should be whether the basolateral amygdala (BLA) is involved in CTA learning through LiCl [11,12,13,14]. To our knowledge, some studies demonstrated that microinjections of the ibotenic acid in the parabrachial nucleus, medial thalamus, and basolateral nucleus of the amygdala may attenuate the formation of LiCl-induced CTA in the acquisition and retention phases, indicating that the BLA contributes to LiCl-induced CTA learning [13,14]. The study of brain imaging has shown that the BLA projects to the NAc, the anterior part of the bed nucleus of the stria terminals, and the central amygdala (CeA) modulates the retrieval process of LiCl-induced CTA learning, indicating the BLA and its pathways contribute to CTA learning in the retrieval phase [15]. Microinjections into the BLA were demonstrated to interfere with remote and recent retrieval effects in an animal model of taste-potentiated odor aversion memory [16]. Another study revealed that when the GABAa receptor agonist muscimol was microinjected into the BLA, it increased the tastant consumption during the retrieval phase. The data indicated that the GABAa receptors were involved in the retrieval process of LiCl-induced CTA learning [17]. In conclusion, these findings were consistent with the viewpoint that the BLA was involved in the CTA learning induced by LiCl. 

However, other studies still needed to support the viewpoint above fully. Another behavioral study used ibotenic acid to destroy the CeA and BLA in the conditioned taste aversion and latent inhibition model to examine the role of the CeA and BLA in CTA learning and latent inhibition. The data showed that in rats with a BLA lesion, it did not affect the taste aversion learning induced by LiCl when the tastant was familiar; moreover, the acquisition of CTA was attenuated when the tastant was novel. However, the CeA lesions did not affect the LiCl-induced CTA in the novel and familiar tastes [12]. Alternatively, after rats were conditioned with LiCl and the tastant to induce the CTA-conditioned learning, the c-Fos data indicated that the nucleus accumbens shell (NAc shell) showed more c-Fos expression; however, the BLA showed less c-Fos expression. The study showed that the NAc shell increased neural activity, but the BLA revealed hypoactivity after LiCl-induced CTA learning [11]. Furthermore, the microinjection of the SCH 23390, a D1 receptor antagonist, in the NAc shell and BLA impaired the conditioned taste suppression. Thus, the D1 receptor (but not the D2 receptor) was involved in LiCl-induced CTA [11]. Therefore, whether the BLA and its D2 receptor are involved in LiCl-induced CTA learning needs further examination. 

Altogether, the present study employed an animal model of LiCl-induced conditioned suppression of the saccharin solution intake to imitate cancer chemotherapy nausea and vomiting symptoms, and thereby, we examined the nature of the role of the BLA, and whether D2 receptors modulate LiCl-induced conditioned suppression of the saccharin solution intake. The study examined whether the microinjection of the sodium chloride blocker lidocaine into the BLA attenuated the conditioned suppression of the saccharin solution induced by LiCl. Moreover, we investigated whether the D2 receptor antagonist haloperidol disrupted the CTA learning through LiCl. Finally, the present data should be discussed, and may provide a novel treatment for ameliorating chemotherapy-induced nausea and vomiting symptoms (see Figure 1). 

## 2. Materials and Methods

### 2.1. Animals

Thirty-seven male Wistar rats (250–350 g) were bought from BioLasco Co., Ltd. (Yi-Lan, Taiwan). All rats were paired-housed in plastic cages (47 cm length × 26 cm width × 21 cm height) with hardwood laboratory bedding (Beta Chip) in a colony room. The colony room was maintained at 20 ± 2 °C and a 12 h light–dark cycle (light on 6:00 AM–6:00 PM) condition. Food and water were available ad libitum, exception for the specific water-deprivation phase. All experiments were performed in compliance with the American Psychological Association ethical standards in the treatment of the animals, or a description of the details of the treatment and received local ethics committee approval. All efforts were made to minimize animal suffering and the number of animals used. 

### 2.2. Microinjections

All of the rats were microinjected with a 0.5 μl volume of vehicle, lidocaine, or haloperidol into both sides of the BLA (anterior/posterior, −2.28 mm from bregma; lateral, ±5.0 mm from the midline; ventral, −8.0 mm from the skull surface) [18]. The injection rate was 0.5 μl/min. The needle was left in the BLA for an additional 1 min. All parameters of the microinjection procedure were referred to in our pilot study. 

### 2.3. Histology: Thionin Staining

The rats were euthanized by pentobarbital overdose and perfused with 0.9% normal saline, followed by a 10% formalin-saline solution. The rats’ brain tissues were removed and placed in a 30% formalin-sucrose solution until fully saturated. The brain was sectioned into 40 μm slices on a freezing microtome, mounted on glass slides, and stained with thionin. The light microscopy observed each slice to verify the location of the injection sites; moreover, the drawings were performed by the brain atlas of Paxinos and Watson (2007). An example of the BLA microinjection using a haloperidol or lidocaine solution is shown in Figure 2A. The serial reconstructions of the largest (gray) and smallest (black) BLA areas are for three coronal levels (−2.16, −2.28, and −2.40 mm posterior to bregma; Figure 2B).

### 2.4. Apparatus

Our pilot study referred to all parameters of surgery and the microinjection procedure. Intake volume was measured in a plastic cage with a single-bottle procedure. The drinking bottle was a 500 ml volume with a metal spout that was connected to a hole in the plastic test cage; the drinking bottle was filled up to 100 ml each trial.

### 2.5. Procedure

In the beginning phase, all animals were allowed to habituate the environment in the home cage for 7 days (Days 1–7). All rats were given the surgery procedure for one day (Day 8). During the surgery procedure, the rat was intraperitoneally injected with atropine sulfate (0.1 mg/rat). Twenty minutes later, the rat was intraperitoneally injected with pentobarbital (50 mg/kg, supplement 5.0 mg/kg). The outer cannula was bilaterally implanted in the BLA. Later, all rats were intraperitoneally injected with gentamicin (6 mg/rat). On Days 9–15, all rats could freely drink water and eat food chow during the recovery phase. Later, all rats were subjected to a water deprivation regimen (23.5 h/day) in their home cage for 1 week (Days 16–22), followed by training to drink with a water bottle in the plastic cage for three days (Days 23–25). During the conditioning phase (Days 26–30), rats were given a 0.1% saccharin solution for 15 min in the plastic test cage and immediately bilaterally injected with a treatment drug into the nucleus of BLA (normal saline or 4% lidocaine solution in Experiment 1; 0.2% tartaric acid solution or 0.5 μg/μl haloperidol in Experiment 2), and then administered 0.15 M LiCl in their home cage. The treatment was given for five sessions. The intake volume of 0.1% saccharin solution was measured for each session. The treatment was completed before noon; after that, water was given for 30 min in the late afternoon (between 1800 and 1900). Thus, all rats were assigned to the Saline (*n* = 11) and 4% Lidocaine (*n* = 8) groups in Experiment 1, and the Vehicle (2% tartaric acid, *n* = 9) or Haloperidol (0.5 μg/μl, *n* = 9) groups in Experiment 2.

### 2.6. Drugs

Saccharin, sodium chloride (NaCl), and lithium chloride (LiCl) were dissolved in distilled water at the following concentrations: 0.1% (*w/v*) saccharin solution, 0.15 M LiCl, and 4% (*w/v*) lidocaine solution were prepared in 0.15 M NaCl. 2% (*w/v*) tartaric acid, and 0.5 μg/μl haloperidol was dissolved into the 2% tartaric acid solution. Injections were intraperitoneal at volumes of 4 mL/kg for LiCl. All chemical compounds were obtained from Sigma Company (Taipei, Taiwan).

### 2.7. Statistical Analysis

A 2 × 5 mixed two-way (group vs. session) analysis of variance (ANOVA) was analyzed for the mean (± SEM) intake volume of 0.1% saccharin solution for five sessions. The mean (± SEM) intake volume (ml) of 0.1% saccharin solution was measured for five consecutive sessions. To avoid individual differences in the intake volume of 0.1% saccharin solution, the intake volume of 0.1 % saccharin solution was transformed into the CTA score (ml) using the following formula: 

Intake volume of 0.1% saccharin solution in the first session—intake volume of 0.1% saccharin solution for each session.

Therefore, a 2 × 4 mixed two-way (group vs. session) ANOVA was performed to analyze the CTA score. When appropriate, the intake volume or CTA score was conducted using a one-way ANOVA.

## 3. Results

### 3.1. Experiment 1: Attenuated Effects of Sodium Ion Channel Blocker Lidocaine in the Basolateral Amygdala to LiCl-Induced CTA Learning

Concerning whether microinjections of sodium ion channel blocker lidocaine in the BLA attenuated LiCl-induced CTA learning, a 5 × 2 mixed two-way ANOVA was performed. The results showed that significant differences occurred between groups [F(1, 17) = 8.69, *p* < 0.05] and sessions [F(4, 68) = 16.79, *p* < 0.05]; nonsignificant differences occurred in group x session [F(4, 68) = 1.98, *p* > 0.05] for 0.1% saccharin solution intake following LiCl administrations. The results indicated that microinjections of sodium ion channel blocker, lidocaine, in the BLA decreased the intake volume of 0.1% saccharin solution (Figure 3). 

### 3.2. Experiment 2: Blunted Effects of D2 Receptor Antagonist Haloperidol in the Basolateral Amygdala to LiCl-Induced CTA Learning

Regarding the issue of whether the BLA’s D2 receptor antagonist haloperidol interfered with LiCl-induced CTA learning, a 2 × 5 mixed two-way ANOVA was conducted for 0.1% saccharin solution following LiCl administrations. The results showed that significant differences occurred in session [F(4, 64) = 20.42, *p* < 0.05] and group x session [F(4, 64) = 7.85, *p* < 0.05]; nonsignificant differences occurred in group [F(1. 16) = 0.17, *p* > 0.05] for 0.1% saccharin solution intake (Figure 4A).

Because the saccharin solution intake in the first session revealed a significant difference between the vehicle and haloperidol groups, the saccharin solution intake was transferred into CTA scores for the vehicle and haloperidol groups. Later, a 2 × 4 mixed two-way ANOVA was performed for CTA scores. The results showed that significant differences occurred in session [F(3, 48) = 14.70, *p* < 0.05] and group x session [F(3, 48) = 11.35, *p* < 0.05] for CTA scores. Moreover, the CTA score of the haloperidol group seemingly significantly increases compared with the vehicle group [F(1, 16) = 3.78, *p* = 0.07]. Therefore, BLA’s microinjections with D2 receptor antagonist haloperidol interfered with LiCl-induced CTA learning (Figure 4B). Note that session 1 in Figure 4B was the first retrieval session and not the first training session. 

## 4. Discussion

The present study showed that lidocaine, a sodium chloride ion channel blocker, could attenuate the saccharin solution suppression, indicating that the LiCl-induced conditioned taste aversion was disrupted by lidocaine in the BLA. Moreover, microinjections of D2 antagonist haloperidol into the BLA blocked conditioned suppression of the saccharin solution intake induced by LiCl. Therefore, the BLA’s sodium chloride channel and D2 receptors modulate LiCl-induced CTA learning. The BLA plays an excitatory role in mediating the LiCl-induced CTA because the BLA lesion disrupted the conditioned suppression of the saccharin solution intake induced by LiCl. Furthermore, D2 receptors of the BLA were involved in LiCl-induced CTA learning.

In summary, the findings indicated that the BLA mechanism in CTA learning went through D2 receptors and sodium chloride ion channels to modulate the conditioned suppression of the saccharin solution intake by LiCl. 

### 4.1. The Basolateral Amygdala and Conditioned Taste Aversion

Concerning the role of the BLA in the conditioned suppression of the tastant intake, previous studies demonstrated that the BLA plays a crucial role in modulating conditioned suppression of tastant induced by LiCl [16,19,20]. For example, the lesion of the BLA with the anodal current reduced the strength of the CTA learning by LiCl [19]. The BLA with lidocaine microinjections disrupted the recent and remote retrieval with taste-potentiated odor aversion memory [16].

Alternatively, previous evidence has suggested that the dopamine system and D2 receptors were involved in LiCl-induced CTA learning. For example, the study of the magnetic resonance imaging approach indicated that the BLA projects to the NAc, the anterior part of the bed nucleus of the stria terminals, and the CeA. These pathways have modulated LiCl’s retrieval of conditioned taste aversion [15]. A previous behavioral study showed that the peripheral injection of the D2 antagonist, haloperidol, between the CS and the US could disrupt LiCl-induced conditioned suppression of the saccharin solution intake. The results indicated that D2 receptors were involved in LiCl-induced CTA learning [21].Besides, the BLA’s GABAa receptors were demonstrated to mediate LiCl-induced CTA learning. For example, a study of behavioral pharmacology showed that the GABAa receptor agonist muscimol’s microinjection into the BLA increased CS consumption, indicating that the GABAa receptor mediated conditioned taste aversion [17]. Therefore, electrical lesions, the D2 receptor antagonist, GABAa receptor antagonist, and sodium ion channel blocker in the BLA could attenuate LiCl’s conditioned suppression of the tastant intake. The BLA and the BLA’s D2 and GABAa receptors modulated LiCl-induced CTA learning. The present data were consistent with the above evidence in behavior.

However, fewer studies did not support this view of the involvement of the BLA in LiCl-induced CTA learning [12,22]. For example, the rats with the BLA lesion did not exhibit different LiCl-induced taste aversion learning with familiar tastes; however, the novel taste, which was paired with LiCl, demonstrated the retardation of CTA learning in the acquisition phase [12]. Another behavioral study showed that the BLA’s DA receptor antagonists (e.g., D1 antagonist SCH 23393 and D2 antagonist raclopride) did not change the latent inhibition of LiCl-induced conditioned taste aversion [22]. 

Overall, the present data were supportive of the previous view that the BLA is involved in CTA induced by LiCl. Moreover, the present data also support that the BLA’s D2 receptors contribute to LiCl-induced CTA learning. 

In addition to the debate about whether the BLA and its D2 receptors are involved in LiCl-induced CTA learning, another line of studies suggested that the BLA was not only involved in the CTA learning by LiCl, but also that the BLA plays an essential role in the various types of conditioned learning [23,24,25,26,27]. For example, the D3 receptors in the BLA were more active in the process of opiate-related reward in the CPP and in withdrawal aversion in CPA memories [27]. Microinjections of the D1 receptor antagonist, SCH 23390, and the D2 receptor antagonist, sulpiride, into the BLA blunted nicotine-induced enhancement of stress-elicited memory impairment in passive avoidance learning; the results indicated that dopamine transmission in the BLA via the D1 or D2 receptor modulated the facilitation effect of nicotine in stress-induced memory retrieval impairment [23]. During the post-training phase, intra-BLA microinfusions of DA could facilitate memory retention; moreover, infusion of the DA receptor antagonist, cis-Flupenthixol, into the NAc shell blocked memory enhancement induced by the intra-BLA infusion of DA. Thus, the interaction of the BLA and NAc shell modulated memory retention in passive avoidance learning [24]; furthermore, the modulation of the BLA for memory retention included noradrenergic and cholinergic systems in the BLA [25]. Another study demonstrated that the 5-HT2 and DA1 receptors in the BLA contributed to conditioned fear and unconditioned fear [26]. In summary, the BLA has a critical role in regulating the different types of conditioned learning besides LiCl-induced CTA learning. This issue needs to be scrutinized further. 

### 4.2. Clinical Implications: Behavioral Interventions for Cancer Chemotherapy-Induced Anticipatory Nausea and Vomiting

Cancer patients with chemotherapy regularly suffer from severe nausea and vomiting symptoms [3,9,10]. Accumulated data have demonstrated that various treatments may prevent the side effects of chemotherapy [28,29,30,31,32]. For example, pharmacological interventions (e.g., benzodiazepines and serotonin-3 receptor antagonists) and nonpharmacological interventions (e.g., behavioral interventions) have been considered for the relief of chemotherapy-induced anticipatory nausea and vomiting [30]. Concerning behavioral interventions, systematic desensitization [28], muscle relaxation training and guided imagery [29], hypnosis [33], overshadowing [31,32], and scapegoating [34,35] have been suggested for the clinical treatment of chemotherapy-induced anticipatory nausea and vomiting. However, there is no evidence of the development of a D2 receptor antagonist of BLA-blunted nausea and vomiting, which results from the taste induced by cancer chemotherapy. This novel treatment ameliorates nausea and vomiting symptoms from the side-effects of chemotherapy, and as a new intervention, should be investigated in further research. 

### 4.3. Emerged Issues and Further Studies

The present data showed that the BLA is essential for controlling LiCl-induced CTA learning. This effect reflected nausea and vomiting in suppressing the saccharin solution intake. However, the CTA learning induced by LiCl may be modulated by a neural network, not only the specific brain areas such as the BLA. Previous studies have suggested that the LiCl-induced conditioned suppression the saccharin solution intake was governed by the NAc, the anterior part of the bed nucleus of the stria terminals, and the central amygdala [15]. Moreover, the BLA and the prelimbic cortex were shown to exhibit more c-Fos expression after morphine- or amphetamine-induced conditioned suppression of the saccharin solution intake [36,37,38,39]. To date, no research has comprehensively examined the issue of whether the dopaminergic pathway or its subtypes of dopamine receptors from the BLA projections to the other brain areas, such as the NAc, VTA, or the subareas of the medial prefrontal cortex (i.e., the cingulate area 1, prelimbic cortex, and infralimbic cortex), actually contribute to the LiCl-induced CTA learning. 

On the other hand, the present study used the emetic drug, LiCl, to induce conditioned suppression of saccharin solution intake. However, another line of behavioral studies has demonstrated that abused drugs (such as morphine [38], cocaine [40], amphetamines [41], and alcohol [42]) could also have the effect of conditioned suppression on the tastant. Until now, no research has examined whether the pathway and dopaminergic receptors of the BLA also mediate abused drug-induced conditioned suppression. Is abused drug-induced suppression different or similar to LiCl-induced conditioned suppression in terms of brain mechanisms? This remains unclear and needs to be investigated further. 

Therefore, these crucial issues should be addressed. The present issues need to be examined using optogenetic or chemogenetic approaches in further research.

### 4.4. Experimental Limitations

The study of the LiCl-induced saccharin solution intake suppression was designed to mimic a cancer chemotherapy-induced nausea and vomiting animal model. However, some experimental limitations should be acknowledged. For example, nausea cannot be assessed, because researchers cannot easily assess this sensation in laboratory animals. Second, standard laboratory animals such as rats and mice are not capable of vomiting, because the rodents do not have a vomiting center similar to the area of the postrema in humans that elicits vomiting behavior. However, the rodents’ area postrema can induce sickness and illness. Therefore, it should be considered whether the CTA learning animal model cannot truly reflect the behaviors of cancer chemotherapy-induced nausea and vomiting. 

## 5. Conclusions

The D2 receptor antagonist (i.e., haloperidol) and sodium ion channel blocker (i.e., lidocaine) attenuated the conditioned suppression of the saccharin solution intake induced by LiCl. The present findings that the blockade of different mechanisms in the BLA interfered with the LiCl’s CTA in nausea and vomiting behaviors could offer some implications and contributions to developing novel treatments for application in the amelioration of cancer chemotherapy-induced side effects such as nausea and vomiting. The present data should be discussed and investigated in further studies.

## Figures and Tables

**Figure 1 brainsci-13-00697-f001:**
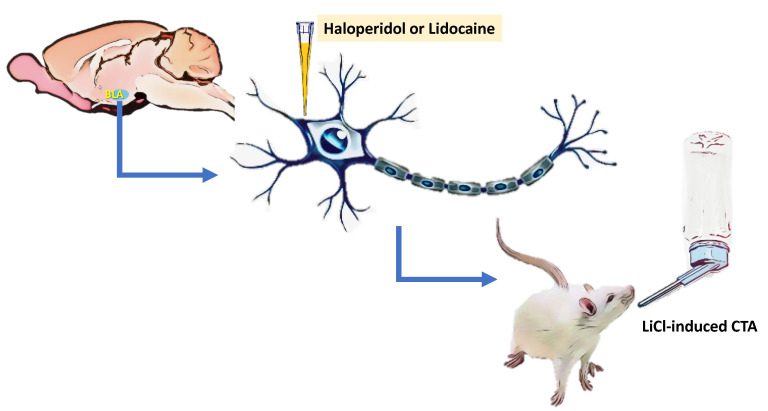
The microinjection of the sodium ion channel blocker, lidocaine, or D2 receptor antagonist, haloperidol into the basolateral amygdala modulates the conditioned suppression of the saccharin solution induced by the emetic drug LiCl.

**Figure 2 brainsci-13-00697-f002:**
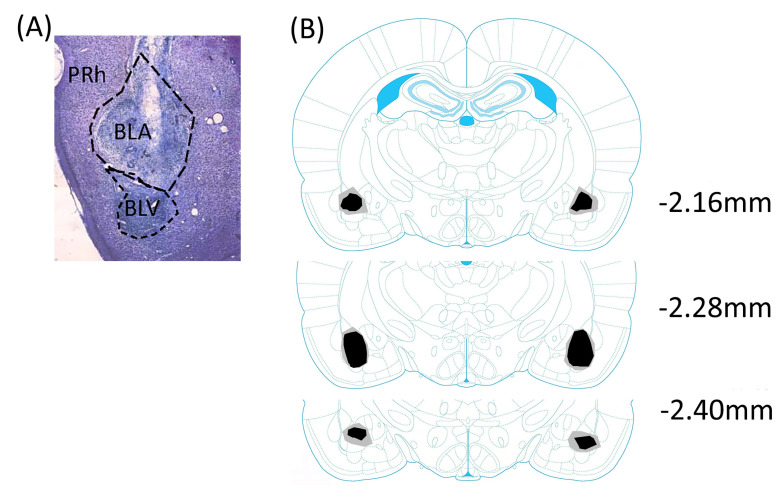
(**A**). Digitalized photomicrographs of thionin-stained coronal sections in the basolateral amygdala. (**B**). Microinjection placements of the largest (gray) and smallest (black) in the basolateral amygdala at three coronal levels (−2.16, −2.28, and −2.40 mm posterior to bregma).

**Figure 3 brainsci-13-00697-f003:**
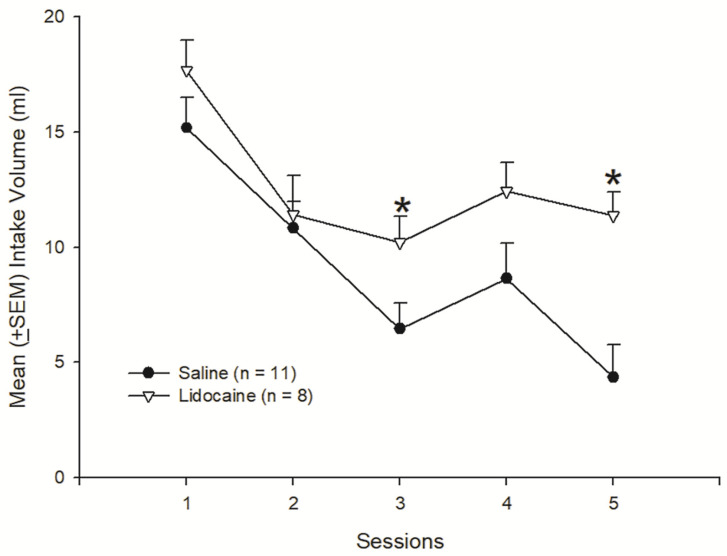
Mean (±SEM) intake volume (mL) of 0.1% saccharin solution for the Saline (*n* = 11) and Lidocaine (*n* = 8) groups. All rats were allowed to freely drink a 0.1 % saccharin solution for 15 min and then received microinjections of the basolateral amygdala with normal saline or 4% lidocaine. Later, rats were given an intraperitoneal injection of a 0.15M LiCl solution. The procedure was one day for a session over 5 sessions. * *p* < 0.05 compared with the Saline group.

**Figure 4 brainsci-13-00697-f004:**
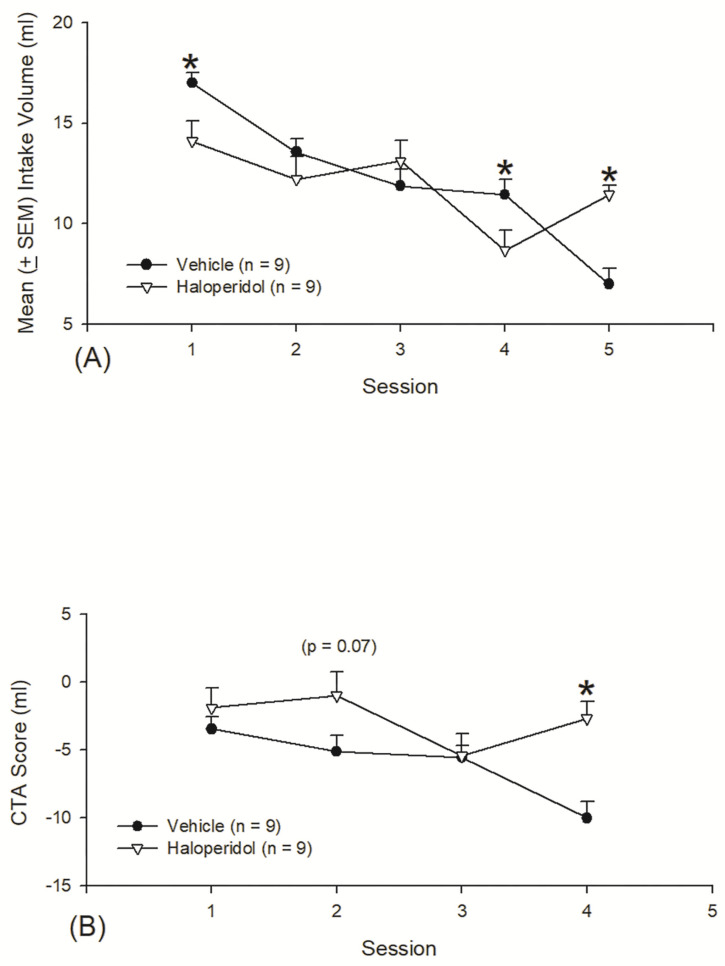
(**A**). Mean (± SEM) intake volume (mL) of 0.1% saccharin solution for the Vehicle (2% tartaric acid, *n* = 9) and Haloperidol (*n* = 9) groups. All rats were allowed to freely drink a 0.1% saccharin solution for 15 min and then received microinjections of the basolateral amygdala with a 2% tartaric acid or haloperidol. Later, rats were given an intraperitoneal injection of a 0.15M LiCl solution. The procedure was one day for a session over 5 sessions. * *p* < 0.05 compared with the Vehicle group. (**B**). CTA scores (mL) for the Vehicle (2% tartaric acid, *n* = 9) and Haloperidol (*n* = 9) groups over 4 sessions. * *p* < 0.05 compared with the Vehicle group.

## Data Availability

The original contributions presented in the study are included in the article, further inquiries can be directed to the corresponding authors.

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
