# Peer review of "D2 Receptors and Sodium Ion Channel Blockades of the Basolateral Amygdala Attenuate Lithium Chloride-Induced Conditioned Taste Aversion Applying to Cancer Chemotherapy Nausea and Vomiting"

_brainsci, 2023, doi:10.3390/brainsci13040697_

Round 1
Reviewer 1 Report
Comments and Suggestions for Authors
This manuscript provides more information related to the involvement of BLA in conditioned taste aversion learning. Authors administered lidocaine or haloperidol before US administration. Conditioned taste aversion learning is delayed after inhibition of electrical or D2 receptors activity within the BLA.
There are some concerns that should be addressed.
Please justify the administration of haloperidol, it is not sufficient to claim that it needs to be examined further. I suggest to amplify information about VTA or other dopaminergic projections to the BLA and their relation to LiCl administration.
Figure 1 may lead to a wrong interpretation. Voltage gated Na+ channels are expressed in the soma and its processes. According to the figure, it seems that channels are expressed within the myelin sheath.
Please explain carefully the used methodology; in 2.2 Surgery procedure section, it seems that drug administration occurred seven days before behavioral protocol. However, section 2.5 indicates that drug administration occurred each training session. The whole manuscript does not describe cannulae implantation, please clarify.
In general, the materials and methods section is too repetitive.
Please check the statistical results in line 170, the degrees of freedom are wrong for session effect. Also, please explain the post hoc analysis used.
CTA score was used for haloperidol experiments, please justify why this was not applied for lidocaine experiments. Also, please correct the graph in figure 4B. Session 1 is the first retrieval session and not the first training session, this may lead to misunderstanding.
Please discuss in depth the involvement of the BLA in CTA learning, there is plenty of evidence indicating its role.
Discuss, thoroughly, the involvement of D2 receptors. These receptors are predominantly presynaptic, thus dopaminergic projections need to be discussed to interpret the obtained data.
Minors:
Line 56, the phrase “…CTA was retarded when…” is misused
Line 21, Please correct the misspelled word VLA to BLA
LIne 58, Please correct “LiCL” to LiCl
The manuscript should be reviewed by a native English speaker.
Author Response
Comments and Suggestions for Authors
Reviewer 1:
This manuscript provides more information related to the involvement of BLA in conditioned taste aversion learning. Authors administered lidocaine or haloperidol before US administration. Conditioned taste aversion learning is delayed after inhibition of electrical or D2 receptors activity within the BLA. There are some concerns that should be addressed.
Point 1: Please justify the administration of haloperidol, it is not sufficient to claim that it needs to be examined further. I suggest to amplify information about VTA or other dopaminergic projections to the BLA and their relation to LiCl administration.
Response 1: Thank you for your comments. We have followed your comments to amplify information about dopaminergic projections from the BLA to the other brain areas to modulate the LiCl-induced CTA learning. Moreover, we have added more studies to justify the BLA contributed to the LiCl’s CTA learning. Please see Lines 42-76.
“…Concerning the studies of chemotherapy nausea and vomiting symptoms, the animal model of LiCl-induced CTA learning and produced CS suppression can be applied to imitate nausea and vomiting behaviors [9-11]. Moreover, a crucial issue should be whether the basolateral amygdala (BLA) was involved in the CTA learning by LiCl [12-15]. To our knowledge, some studies demonstrated that the microinjections of the ibotenic acid in the parabrachial nucleus, medial thalamus, and basolateral nucleus of the amygdala could attenuate the formation of LiCl-induced CTA in the acquisition and retention phases, indicating that the BLA contributed to the LiCl-induced CTA learning [14,15]. The study of the brain imaging showed that the BLA projects to the NAc, anterior part of the bed nucleus of the stria terminals, and the central amygdala (CeA) modulated the retrieval process of the LiCl-induced CTA learning, indicating the BLA and its pathways contributed to the CTA learning in the retrieval phase [16]. Microinjections of the BLA were demonstrated to interfere with remote and recent re-trieval effects in the animal model of taste-potentiated odor aversion memory [17]. Another study revealed that when GABAa receptor agonist muscimol was microinjected into the BLA, it increased the tastant consumption during the retrieval phase. The data indicated that the GABAa receptors were involved in the retrieval process of LiCl-induced CTA learning [18]. In conclusion, these findings were consistent with the viewpoint that the BLA was involved in the CTA learning induced by LiCl.
However, other studies still needed to support the viewpoint above fully. Another behavioral study used ibotenic acid to destroy the CeA and BLA on the conditioned taste aversion and latent inhibition model to examine the role of the CeA and BLA in CTA learning and latent inhibition. The data showed that rats with the BLA lesion did not affect taste aversion learning induced by LiCl when the tastant is familiar; moreover, the acquisition of CTA was attenuated when the tastant was novel; however, the CeA lesions did not affect the LiCl-induced CTA in the novel and familiar tastes [13]. Alternatively, the c-Fos labeling study indicated that, following the LiCl-induced CTA learning, the nucleus accumbens shell (NAc shell) were more c-Fos expression; how-ever, the BLA showed less c-Fos expression, indicating the NAc shell increased neural activity, but the BLA revealed hypoactivity after LiCl-induced CTA learning [12]. Furthermore, the microinjection of the SCH 23390, D1 receptor antagonist, in the NAc shell and BLA impaired the conditioned taste suppression. Thus, D1 receptor (but not the D2 receptor) was involved in LiCl-induced CTA [12]. Therefore, the issue of what role the BLA modulates LiCl-induced CTA learning and whether the D2 receptor was also involved in LiCl-induced CTA learning needs to be examined further.…”
Point 2: Figure 1 may lead to a wrong interpretation. Voltage gated Na+ channels are expressed in the soma and its processes. According to the figure, it seems that channels are expressed within the myelin sheath.
Response 2: Thank you for your comments. We have revise Figure 1. Please check it again.
Point 3: Please explain carefully the used methodology; in 2.2 Surgery procedure section, it seems that drug administration occurred seven days before behavioral protocol. However, section 2.5 indicates that drug administration occurred each training session. The whole manuscript does not describe cannulae implantation, please clarify.
Response 3: Thank you for your comments. We have followed your comments to add the description related to the cannula implantation and modified the contents of 2.5. Procedure. Please see Lines 128-147.
“…2.5. Procedure
In the beginning phase, all animals were allowed to habituate the environment in the home cage for 7 days (Days 1-7). All rats were given the surgery procedure for one day (Day 8). During the surgery procedure, the rat was intraperitoneally injected with atropine sulfate (0.1 mg/rat). Twenty mines later, the rat was intraperitoneally injected with pentobarbital (50 mg/kg, supplement 5.0 mg/kg). The outer cannula was bilateral-ly implanted in the BLA. Later, all rats have intraperitoneally injected with gentamicin (6 mg/rat). On Days 9-15, all rats could freely drink water and eat food chow during the recovery phase. Later, all rats were subjected to a water deprivation regimen (23.5 hr/day) in their home cage for 1 week (Days 16-22), followed by training to drink with a water bottle in the plastic cage for three days (Days 23-25). During the conditioning phase (Days 26-30), rats were given a 0.1% sucrose solution for 15 min in the plastic test cage and immediately bilaterally injected with a treatment drug into the nucleus of BLA (normal saline or 4% lidocaine solution in Experiment 1; 0.2 % tartaric acid solu-tion or 0.2 mg/kg haloperidol in Experiment 2), and then administered 0.15 M LiCl in their home cage. The treatment was given for five sessions. The intake volume of 0.1% saccharin solution was measured for each session. The treatment was completed be-fore noon; after that, water was given for 30 min in the late afternoon (between 1800 and 1900). Thus, all rats were assigned to the Saline (n = 11) and 4% Lidocaine (n = 8) groups in Experiment 1, and the Vehicle (2% tartaric acid, n = 9) or Haloperidol (0.2 mg/kg, n = 9) in Experiment 2…”
Point 4: In general, the materials and methods section is too repetitive.
Response 4: Thank you for your comment. We have revised this point and deleted the repetitive sentences in the Materials and Methods section. Please see the whole contents of the Materials and Methods section.
Point 5: Please check the statistical results in line 170, the degrees of freedom are wrong for session effect. Also, please explain the post hoc analysis used.
Response 5: Thank you for your comment. We have revised the degrees of freedom and F-value for session on Lines 174. Based on the rules of statistical analysis, the factor of group was only two levels for each session. Thus, the post hoc analysis cannot be performed.
Point 6: CTA score was used for haloperidol experiments, please justify why this was not applied for lidocaine experiments. Also, please correct the graph in figure 4B. Session 1 is the first retrieval session and not the first training session, this may lead to misunderstanding.
Response 6: Thank you for your comments. Because the intake volume of saccharin solution in the first session was not significant differences between the Saline and Lidocaine groups, it did not need to transfer the intake volume of saccharin solution into the CTA score.
Also, we have added the sentence to clarify the session 1 in Figure 4B is the first retrieval session and not the first training session. Please see Lines 201-202.
“…Note that session 1 in Figure 4B was the first retrieval session and not the first training session…”
Point 7: Please discuss in depth the involvement of the BLA in CTA learning, there is plenty of evidence indicating its role.
Response 7: Thank you for your comments. We have discussed the involvement of BLA in CTA learning. Please see Lines 223-270.
“…4.1. The basolateral amygdala, dopamine system, and conditioned taste aversion
Concerning the role of the BLA in the conditioned suppression of the tastant in-take, previous studies demonstrated that the BLA plays a crucial role in modulating conditioned suppression of tastant induced by LiCl [17,20,21]. For example, the study of the magnetic resonance imaging approach indicated that the BLA projects to the NAc, the anterior part of the bed nucleus of the stria terminals, and the CeA. These pathways have modulated LiCl's retrieval of conditioned taste aversion [16]. The pre-vious behavioral study showed that the peripheral injection of the D2 antagonist, haloperidol, between the CS and the US could disrupt LiCl-induced conditioned sup-pression of the saccharin solution intake. The results indicated that D2 receptors were involved in LiCl-induced CTA learning [22].
These data indicated that the BLA's GABAa receptor and sodium ion channels were involved in LiCl-induced CTA learning. Moreover, the GABAa receptor agonist muscimol microinjection into the BLA increased CS consumption, indicating that the GABAa receptor mediated the conditioned taste aversion [18]. The lesion of the BLA with the anodal current reduced the strength of the CTA learning by LiCl [20]. The BLA with lidocaine microinjections disrupted the recent and remote retrieval with taste-potentiated odor aversion memory [17]. Therefore, GABAa receptor antagonist, sodium ion channel blocker, and electrical lesions in the BLA could attenuate LiCl's conditioned suppression of the tastant intake.
However, fewer studies did not support this view of the involvement of the BLA in LiCl-induced CTA learning [13,23]. For example, the rats with the BLA lesion did not affect LiCl-induced taste aversion learning in the familiar taste; however, the novel taste, which was paired with LiCl, demonstrated the retardation of CTA learning in the acquisition phase [13]. Another behavioral study showed that the BLA DA receptor antagonists (e.g., D1 antagonist SCH 23393 and D2 antagonist raclopride) did not change the latent inhibition of LiCl-induced conditioned taste aversion [23]. Therefore, the present data were consistent with the previous view that the BLA was involved in CTA by LiCl. Furthermore, the present data indicated that the BLA’s D2 receptors and sodium ion channels contributed to LiCl-induced CTA learning.
Alternatively, the BLA was not only involved in the CTA learning by LiCl, but al-so the BLA might play an essential role in the various types of conditioned learning [24-28]. For example, the D3 receptors in the BLA were more active in the process of opiate-related reward in the CPP and withdrawal aversion in the CPA memories [28]. Microinjections of the D1 receptor antagonist, SCH 23390, and the D2 receptor antago-nist, sulpiride, into the BLA blunted nicotine-induced enhancement of stress-elicited memory impairment in the passive avoidance learning; the results indicated that do-pamine transmission in the BLA via the D1 or D2 receptor modulated the facilitation effect of nicotine in stress-induced memory retrieval impairment [24]. During the post-training phase, intra-BLA microinfusions of DA could facilitate memory retention; moreover, infusion of DA receptor antagonist, cis-Flupenthixol, into the NAc shell blocked memory enhancement induced by the intra-BLA infusion of DA. Thus, the in-teraction of the BLA and NAc shell modulated memory retention in passive avoidance learning [25]; furthermore, the modulation of the BLA for memory retention included noradrenergic and cholinergic systems in the BLA [26]. Another study demonstrated that the 5-HT2 and DA1 receptors in the BLA contributed to conditioned fear and un-conditioned fear [27]. In summary, the BLA has a critical role in regulating the differ-ent types of conditioned learning besides LiCl-induced CTA learning...”
Point 8: Discuss, thoroughly, the involvement of D2 receptors. These receptors are predominantly presynaptic, thus dopaminergic projections need to be discussed to interpret the obtained data.
Response 8: Thank you or your comments. Because no research comprehensively examined the issue of the involvement of D2 receptors and its dopaminergic projections from the BLA. Thus, we have discussed and added this issue in 4.3. Emerged issues and further studies on Lines 288-312.
“…4.3. Emerged issues and further studies
The present data showed that the BLA is essential for controlling LiCl-induced CTA learning. This effect reflected nausea and vomiting in suppressing the saccharin solution intake. However, the CTA learning induced by LiCl may be modulated by a neural network, not only the specific brain areas such as the BLA. Previous studies suggested that the LiCl-induced conditioned suppression of the saccharin solution in-take was governed by the NAc, the anterior part of the bed nucleus of the stria termi-nals, and the central amygdala [16]. Moreover, the BLA and the prelimbic cortex were shown to be more c-Fos expression after morphine- or amphetamine-induced condi-tioned suppression of the saccharin solution intake [37-40]. To date, no research com-prehensively examined the issue of whether the dopaminergic pathway or its subtypes of dopamine receptors from the BLA projections to the other brain areas such as the NAc, VTA, or the subareas of the medial prefrontal cortex (i.e., the cingulate area 1, prelimbic cortex, and infralimbic cortex) contributed to the LiCl-induced CTA learning.
On the other hand, the present study used the emetic drug, LiCl, to induce conditioned suppression of the saccharin solution intake. However, another line of behavioral studies demonstrated that the abused drugs (such as morphine [39], cocaine [41], amphetamine [42], and alcohol [43]) could also induce the conditioned suppression effect for the tastant. Until now, no research examined whether the pathway and dopaminergic receptors of the BLA also mediate abused drug-induced conditioned sup-pression. Was the abused drugs induced suppression different or similar to LiCl-induced conditioned suppression in brain mechanisms? It remains unclear, and it needs to be investigated further.
Therefore, these crucial issues should be addressed. The present issues need to be examined using optogenetic or chemogenetic approaches in further research...”
Minors:
Point 9: Line 56, the phrase “…CTA was retarded when…” is misused
Response 9: Thank you for your comments. We have revised this sentence. Please see Lines 65-66.
“…moreover, the acquisition of CTA was attenuated when the tastant is novel…”
Point 10: Line 21, Please correct the misspelled word VLA to BLA
Response 10: Thank you for your comment. We have corrected it. Please see Line 20.
“…The present results showed that lidocaine microinjections into the BLA could attenuate the LiCl-induced CTA…”
Point 11: Line 58, Please correct “LiCL” to LiCl
Response 11: Thank you for your comments. We have revised it. Please see Line 58.
Point 12: The manuscript should be reviewed by a native English speaker.
Response 12: Thank you for your comments. We have followed your comments to reviewed by a native English speaker.
Reviewer 2 Report
Comments and Suggestions for Authors
The manuscript “D2 receptors and sodium ion channel blockades of the basolateral amygdala attenuate lithium chloride-induced conditioned taste aversion applying to cancer chemotherapy nausea and 4 vomiting” by Gao et al is a research article which examined the effects of D2 receptors and sodium ion channel blockades of the basolateral amygdala on lithium chloride-induced conditioned taste aversion which imitates the chemotherapy nausea and vomiting. The authors found that lidocaine or D2 blocker haloperidol injection into the basolateral amygdala attenuated the lithium chloride-induced conditioned taste aversion. Therefore, the authors concluded that the blockade of D2 receptors and sodium ion channels in the basolateral amygdala would attenuate cancer chemotherapy-induced nausea and vomiting. Generally, the subject is of interest and scientifically sound and contains essential contents. This paper is also of importance for providing us the important evidence that the D2 receptors and sodium ion channels in the basolateral amygdala are involved in lithium chloride-induced conditioned taste aversion. The manuscript has been well organized and written. However, I have some concerns on the paper.
1. Abstract: “VLA” should be changed to ”BLA”.
2. In the scatter plots, all the data plots should be displayed if possible. The readers can obtain more information from these data.
3. In this study, CTA learning induced by LiCl is used to mimic the cancer chemotherapy-induced nausea and vomiting animal model. This is because creating an animal model of nausea and vomiting has severe limitations. The first limitation is that nausea cannot be assessed because researchers cannot easily assess this sensation in laboratory animals. The another limitation is that standard laboratory animals such as rat and mice are not capable of vomiting because the rodents do not have a vomiting center. However, the CTA learning animal model cannot truly reflect the behaviors of cancer chemotherapy-induced nausea and vomiting. Please add the limitation of the study.
Author Response
Comments and Suggestions for Authors
Reviewer 2:
The manuscript “D2 receptors and sodium ion channel blockades of the basolateral amygdala attenuate lithium chloride-induced conditioned taste aversion applying to cancer chemotherapy nausea and 4 vomiting” by Gao et al is a research article which examined the effects of D2 receptors and sodium ion channel blockades of the basolateral amygdala on lithium chloride-induced conditioned taste aversion which imitates the chemotherapy nausea and vomiting. The authors found that lidocaine or D2 blocker haloperidol injection into the basolateral amygdala attenuated the lithium chloride-induced conditioned taste aversion. Therefore, the authors concluded that the blockade of D2 receptors and sodium ion channels in the basolateral amygdala would attenuate cancer chemotherapy-induced nausea and vomiting. Generally, the subject is of interest and scientifically sound and contains essential contents. This paper is also of importance for providing us the important evidence that the D2 receptors and sodium ion channels in the basolateral amygdala are involved in lithium chloride-induced conditioned taste aversion. The manuscript has been well organized and written. However, I have some concerns on the paper.
Point 1: Abstract: “VLA” should be changed to ”BLA”.
Response 1: Thank you for your comments. We have revised it. Please see Lines 20.
Point 2: In the scatter plots, all the data plots should be displayed if possible. The readers can obtain more information from these data.
Response 2: Thank you for your comments. We have shown all information of all the data. For example, Figure 3 showed the mean of saccharin solution intake after the LiCl conditioning over five sessions. Figure 4A also showed that the mean of saccharin solution intake after LiCl’s conditioning for five sessions. Due to the individual differences for the first session without LiCl’s conditioning effects, we transferred the raw data of the saccharin solution intake into the CTA scores in Figure 4B. Thus, the Figure 4B only showed four sessions. We have already tried to show all information of the raw data in Figures 3 and 4.
Point 3: In this study, CTA learning induced by LiCl is used to mimic the cancer chemotherapy-induced nausea and vomiting animal model. This is because creating an animal model of nausea and vomiting has severe limitations. The first limitation is that nausea cannot be assessed because researchers cannot easily assess this sensation in laboratory animals. The another limitation is that standard laboratory animals such as rat and mice are not capable of vomiting because the rodents do not have a vomiting center. However, the CTA learning animal model cannot truly reflect the behaviors of cancer chemotherapy-induced nausea and vomiting. Please add the limitation of the study.
Response 3: Thank you for your comments. We have followed your comments to add the experimental limitation section in Discussion. Please see the experimental limitation on Lines 314-323.
“…4.4. Experimental limitations
The study of the LiCl-induced saccharin solution intake suppression was designed to mimic the cancer chemotherapy-induced nausea and vomiting animal model. How-ever, some experimental limitations should be concerned. For example, nausea cannot be assessed because researchers cannot easily assess this sensation in laboratory animals. Second, standard laboratory animals such as rats and mice are not capable of vomiting because the rodents do not have a vomiting center like the human’s area postrema that elicits the vomiting behavior. However, the rodents’ area postrema can induce sickness and illness. Therefore, it should be concerned whether the CTA learning animal model cannot truly reflect the behaviors of cancer chemotherapy-induced nausea and vomiting…”
Round 2
Reviewer 1 Report
Comments and Suggestions for Authors
The authors improved the manuscript. Nevertheless there are still some concerns.
Line 18 Please correct LiCL to LiCl
Lines 68-69 the sentence requires editing
Lines 75-76 need to be corrected
Lines 83-84, I suggest to rephrase this sentence, please check
Line 132, where it says "mines" it should say minutes
Line 139, I think there is a mistake in the solution, animals drank saccharin throughout the experiments.
Authors mentioned that haloperidol (0.2 mg/kg) was administered. However this is an unusual unit for intraparenchymal administrations. Please clarify
Author Response
Comments and Suggestions for Authors
Reviewer 1:
The authors improved the manuscript. Nevertheless there are still some concerns.
Point 1: Line 18 Please correct LiCL to LiCl.
Response 1: Thank you for your comments. We have followed your comments to revise this point. Please see Line 18.
Point 2: Lines 68-69 the sentence requires editing
Response 2: Thank you for your comments. We have followed your comments to revise this sentence. Please see Lines 67-72.
“…Alternatively, after rats were conditioned with LiCl and the tastant to induce the CTA conditioned learning, the c-Fos data indicated that the nucleus accumbens shell (NAc shell) were more c-Fos expression; however, the BLA showed less c-Fos expression. The study showed that the NAc shell increased neural activity, but the BLA revealed hypoactivity after LiCl-induced CTA learning [12]…”
Point 3: Lines 75-76 need to be corrected
Response 3: Thank you for your comments. We have corrected this sentence on Lines 75-76. Please check it again.
“…Therefore, whether the BLA and its D2 receptor were involved in LiCl-induced CTA learning needs further examination…”
Point 4: Lines 83-84, I suggest to rephrase this sentence, please check.
Response 4: Thank you for your comments. We have followed your comments to rephrase this sentence on Lines 83-84. Please see Lines 83-84.
“…Moreover, whether the D2 receptor antagonist haloperidol disrupted the CTA learning by LiCl?...”
Point 5: Line 132, where it says "mines" it should say minutes
Response 5: Thank you for your comments. We have corrected it on Line 132.
“…Twenty minutes later, the rat was intraperitoneally injected with pentobarbital (50 mg/kg, supplement 5.0 mg/kg)…”
Point 6: Line 139, I think there is a mistake in the solution, animals drank saccharin throughout the experiments.
Response 6: Thank you for your comments. We have corrected this sentence. Please check Line 139.
“…During the conditioning phase (Days 26-30), rats were given a 0.1% saccharin solution for 15 min in the plastic test cage…”
Point 7: Authors mentioned that haloperidol (0.2 mg/kg) was administered. However this is an unusual unit for intraparenchymal administrations. Please clarify
Response 7: Thank you for your comments. We have followed your comments to revise this point. We found that the correct concentration was 0.5 mg/ml after checking our data. Please see Lines 138-154.
“…During the conditioning phase (Days 26-30), rats were given a 0.1% saccharin solution for 15 min in the plastic test cage and immediately bilaterally injected with a treatment drug into the nucleus of BLA (normal saline or 4% lidocaine solution in Experiment 1; 0.2 % tartaric acid solution or 0.5 mg/ml haloperidol in Experiment 2), and then administered 0.15 M LiCl in their home cage. The treatment was given for five sessions. The intake volume of 0.1% saccharin solution was measured for each session. The treatment was completed before noon; after that, water was given for 30 min in the late afternoon (between 1800 and 1900). Thus, all rats were assigned to the Saline (n = 11) and 4% Lidocaine (n = 8) groups in Experiment 1, and the Vehicle (2% tartaric acid, n = 9) or Haloperidol (0.5 mg/ml, n = 9) in Experiment 2.
2.6. Drugs
Saccharin, sodium chloride (NaCl), and lithium chloride (LiCl) were dissolved in distilled water at the following concentrations: 0.1% (w/v) saccharin solution, 0.15 M LiCl, and 4 % (w/v) lidocaine solution was prepared in the 0.15 M NaCl. 2% (w/v) tar-taric acid and 0.5 mg/ml haloperidol were dissolved into the 2 % tartaric acid solution. Injections were intraperitoneal at volumes of 4 ml/kg for LiCl. All chemical compounds were obtained from Sigma Company (Taipei, Taiwan)…”
Round 3
Reviewer 1 Report
Comments and Suggestions for Authors
I am writing you to recommend the acceptance of Gaos's paper tilted "D2 receptors and sodium ion channel blockades of the basolateral amygdala attenuate lithium chloride-induced conditioned 3 taste aversion applying to cancer chemotherapy nausea and vomiting" for publication in Brain Sciences. While the content of the manuscript is sufficient to be published, there are several grammatical errors and awkward phrasings throughout the text that could be addressed with a professional edit. Moreover, I still do have some doubts about both the data and discussion sections of the paper. While I recognize the importance of the data, I feel that it is borderline in terms of the quality and quantity required for publication in "Brain Sciences"